# Discontinuation of BRAF/MEK-Directed Targeted Therapy after Complete Remission of Metastatic Melanoma—A Retrospective Multicenter ADOReg Study

**DOI:** 10.3390/cancers13102312

**Published:** 2021-05-12

**Authors:** Henner Stege, Maximilian Haist, Michael Schultheis, Maria Isabel Fleischer, Peter Mohr, Friedegund Meier, Dirk Schadendorf, Selma Ugurel, Elisabeth Livingstone, Lisa Zimmer, Rudolf Herbst, Claudia Pföhler, Katharina Kähler, Michael Weichenthal, Patrick Terheyden, Dorothée Nashan, Dirk Debus, Martin Kaatz, Fabian Ziller, Sebastian Haferkamp, Andrea Forschner, Ulrike Leiter, Alexander Kreuter, Jens Ulrich, Johannes Kleemann, Fabienne Bradfisch, Stephan Grabbe, Carmen Loquai

**Affiliations:** 1Department of Dermatology, University Medical Center Mainz, 55131 Mainz, Germany; Henner.Stege@unimedizin-mainz.de (H.S.); Michael.Schultheis@unimedizin-mainz.de (M.S.); mariaisabel.fleischer@unimedizin-mainz.de (M.I.F.); Stephan.Grabbe@unimedizin-mainz.de (S.G.); carmen.loquai@unimedizin-mainz.de (C.L.); 2Department of Dermatology, Elbe Kliniken Buxtehude, 21614 Buxtehude, Germany; Peter.Mohr@elbekliniken.de; 3Skin Cancer Center at the University Cancer Center Dresden and National Center for Tumor Diseases, 01307 Dresden, Germany; Friedegund.Meier@uniklinikum-dresden.de; 4Department of Dermatology, University Hospital Carl Gustav Carus, TU Dresden, 01307 Dresden, Germany; 5Department of Dermatology, University Hospital Essen, 45122 Essen, Germany; Dirk.schadendorf@uk-essen.de (D.S.); selma.ugurel@uk-essen.de (S.U.); Elisabeth.Livingstone@uk-essen.de (E.L.); Lisa.Zimmer@uk-essen.de (L.Z.); 6Department of Dermatology, Helios-Hospital Erfurt, 99089 Erfurt, Germany; rudolf.herbst@helios-gesundheit.de; 7Department of Dermatology, Saarland University Medical Center, 66421 Homburg/Saar, Germany; claudia.pfoehler@uks.eu; 8Department of Dermatology, Campus Kiel, University Hospital of Schleswig-Holstein Hospital, 24105 Kiel, Germany; kkaehler@dermatology.uni-kiel.de (K.K.); MWeichenthal@dermatology.uni-kiel.de (M.W.); 9Department of Dermatology, Allergology, and Venereology, University of Lübeck, 23538 Lübeck, Germany; patrick.terheyden@uksh.de; 10Department of Dermatology, Hospital Dortmund, 44137 Dortmund, Germany; Dorothee.Nashan@klinikumdo.de; 11Department of Dermatology, Paracelsus Medical University Nuremberg, City Hospital of Nuremberg, 90408 Nuremberg, Germany; dirk.debus@klinikum-nuernberg.de; 12Department of Dermatology, Wald-Klinikum Gera, 07548 Gera, Germany; Martin.Kaatz@srh.de; 13Department of Dermatology, DRK Krankenhaus Rabenstein, 09117 Chemnitz, Germany; ziller.fabian@drk-khs.de; 14Department of Dermatology, University Hospital Regensburg, 93053 Regensburg, Germany; Sebastian.Haferkamp@klinik.uni-regensburg.de; 15Department of Dermatology, University Hospital Tübingen, 72076 Tübingen, Germany; andrea.forschner@med.uni-tuebingen.de (A.F.); Ulrike.Leiter@med.uni-tuebingen.de (U.L.); 16Department of Dermatology, Venereology and Allergology, Helios St. Elisabeth Klinik Oberhausen, University Witten-Herdecke, 46045 Oberhausen, Germany; alexander.kreuter@helios-gesundheit.de; 17Department of Dermatology, Klinikum Quedlinburg, 06484 Quedlinburg, Germany; Jens.Ulrich@harzklinikum.com; 18Department of Dermatology, Venereology and Allergy, Johann Wolfgang Goethe University, 60590 Frankfurt/Main, Germany; Johannes.Kleemann@kgu.de (J.K.); Fabienne.Bradfisch@kgu.de (F.B.)

**Keywords:** targeted therapy, complete response, advanced melanoma, discontinuation, disease progression, second-line immunotherapy

## Abstract

**Simple Summary:**

The introduction of BRAF/MEK-directed targeted therapy (TT) has significantly improved the management of patients with advanced BRAF-V600-mutant melanoma. Although resistance occurs, there is a subgroup of patients showing a complete response (CR) to TT and who maintain durable disease control. For these patients with durable CR, it is not clear whether it is safe to cease therapy. In this retrospective, multicenter study we have analyzed 37 patients who received TT and achieved a CR upon treatment. We identified 15 patients with a durable CR to TT. Overall, patients who discontinued TT (n = 26) were at higher risk of tumor progression compared to patients receiving ongoing TT. Sustained CR was however not restricted to patients with ongoing TT (n = 11) but was also found in patients who ceased TT (n = 4). Finally, our analysis indicated which patients with an initial CR might be most likely to maintain durable CR upon discontinuation of TT.

**Abstract:**

The advent of BRAF/MEK inhibitors (BRAFi/MEKi) has significantly improved progression-free (PFS) and overall survival (OS) for patients with advanced BRAF-V600-mutant melanoma. Long-term survivors have been identified particularly among patients with a complete response (CR) to BRAF/MEK-directed targeted therapy (TT). However, it remains unclear which patients who achieved a CR maintain a durable response and whether treatment cessation might be a safe option in these patients. Therefore, this study investigated the impact of treatment cessation on the clinical course of patients with a CR upon BRAF/MEK-directed-TT. We retrospectively selected patients with BRAF-V600-mutant advanced non-resectable melanoma who had been treated with BRAFi ± MEKi therapy and achieved a CR upon treatment out of the multicentric skin cancer registry ADOReg. Data on baseline patient characteristics, duration of TT, treatment cessation, tumor progression (TP) and response to second-line treatments were collected and analyzed. Of 461 patients who received BRAF/MEK-directed TT 37 achieved a CR. TP after initial CR was observed in 22 patients (60%) mainly affecting patients who discontinued TT (n = 22/26), whereas all patients with ongoing TT (n = 11) maintained their CR. Accordingly, patients who discontinued TT had a higher risk of TP compared to patients with ongoing treatment (*p* < 0.001). However, our data also show that patients who received TT for more than 16 months and who discontinued TT for other reasons than TP or toxicity did not have a shorter PFS compared to patients with ongoing treatment. Response rates to second-line treatment being initiated in 21 patients, varied between 27% for immune-checkpoint inhibitors (ICI) and 60% for BRAFi/MEKi rechallenge. In summary, we identified a considerable number of patients who achieved a CR upon BRAF/MEK-directed TT in this contemporary real-world cohort of patients with BRAF-V600-mutant melanoma. Sustained PFS was not restricted to ongoing TT but was also found in patients who discontinued TT.

## 1. Introduction

The advent of mitogen-activated protein kinase (MAPK) pathway inhibitors [1,2,3,4,5] and immune checkpoint inhibitors (ICI) [6,7,8] has dramatically changed the treatment landscape for patients with advanced melanoma. Prior to the introduction of these classes of medication, chemotherapy represented the only treatment option for patients with advanced melanoma and disease outcomes were poor with a median overall survival (OS) of approximately 7.5 months and a 5-year survival rate of less than 10% [3,9,10]. Nearly 50% of cutaneous melanomas harbor a BRAF-V600 mutation, resulting in a constitutive activation of the MAPK pathway [11,12]. Through direct inhibition of the BRAF-V600 driver mutation via BRAF inhibitors (BRAFi) a prolonged progression-free and overall survival have been overserved [5,13]. Nevertheless, disease progression after a median progression-free survival of 6–7 months was a common theme in therapy with single-agent BRAFi [1,4]. Reactivation of the MAPK pathway, development of de novo NRAS or MEK mutations, or variant splicing of mutant BRAF-V600 are among the mechanisms associated with the development of secondary acquired resistance [14,15,16]. The combination of a BRAFi with a MEK inhibitor (MEKi) has resulted in a significantly improved PFS and OS as well as a decreased incidence of adverse events [3,17,18]. The combinations of BRAF and MEK inhibitors have demonstrated immediate anti-tumor effects, which culminate in tumor regression and symptomatic improvement. However, response durability and the degree of response differ: Emerging data from a recently published landmark analysis suggests that a subset of patients with a CR to combination BRAFi/MEKi treatment shows a profound and durable response with a 5-year overall survival rate of 71% [19,20]. In contrast, patients with stable disease (SD) as the best overall response (BOR) only have a 5-year OS rate of 16%. Overall, partial response was reported in approximately 50% of the patients, whereas complete response, on the other hand, is recorded in 3–6% of the patients [4,21]. 

Unlike many ICI studies, where treatment is often only continued for a finite period of time, BRAFi/MEKi studies are regularly continued indefinitely, irrespective of progression or unacceptable toxicities [22]. Although TT protocols have generally a good tolerability, adverse events can occur frequently and represent one of the main reasons for treatment discontinuation [23,24]. Taking into account the clinical impact of persisting low-grade toxicities on the patients’ quality of life and the potential risk of secondary malignancies [25,26], it remains a pending issue to determine whether BRAFi and MEKi should be continued until progression or whether it can be stopped at an earlier time point without posing a significant risk for subsequent tumor progression (TP). 

In particular, it is currently unclear how long BRAFi and MEKi should be continued for patients with an initial CR and if treatment cessation might impact the clinical outcome of these patients. Investigating the clinical course of patients with an initial CR to BRAF/MEK-directed TT might further help to identify potential factors affecting the duration of response and thus optimize individual treatment strategies after CR to TT. 

In the current study, we retrospectively examined a cohort of patients with BRAF-V600-mutant metastatic melanoma who were treated with BRAFi/MEKi and achieved an initial CR upon treatment. Data on the presented patient cohort were collected within the framework of the “Registery of the Arbeitsgemeinschaft Dermatologische Onkologie” (ADOReg). In order to assess the outcome of patients with CR after cessation of TT, we have compared three subgroups of patients: Patients with CR, who did continuously receive BRAF/MEK-directed TT upon CR, patients who had to discontinue TT due to TP or toxicity, and patients who did discontinue TT for other reasons, such as the personal wish of the patient or at the recommendation of the supervising physician (i.e., long-term moderate toxicities). Additionally, we aimed to identify potential factors predicting the clinical outcome of patients after treatment discontinuation, such as clinicopathological factors or the initial duration of TT. Last, we investigated the clinical response of patients who received ICI therapy as second-line treatment subsequent to TP during BRAF/MEK-directed TT. Such investigations are clinically relevant given the possibility of disease progression after cessation of BRAF/MEK-directed TT and the subsequent exigency to efficiently rechallenge tumor treatment. 

## 2. Patients and Methods

We retrospectively analyzed all patients with BRAF-V600 advanced melanoma and CR upon BRAF-MEK-directed TT, who were included into the multicentric skin cancer registry ADOReg of the Dermatologic Cooperative Oncology Group (DeCOG) until the data cut-off (04/2020). 

Data on the age at the start of TT with BRAFi and MEKi, sex, primary localization of melanoma, site of metastasis, LDH levels in serum, systemic pretreatments, duration of treatment, BOR to treatment, duration of the initial response, overall status of the patient, reasons for discontinuation, subsequent course of the disease and second-line treatment were collected. BOR was defined as complete response (CR), partial response (PR), stable disease (SD), or progressive disease (PD). PD was defined by disease recurrence at any site during observation according to standard RECIST criteria. In this retrospective analysis we included patients with advanced non-resectable melanoma (n = 2441/5231, 46.7%) who received first-line BRAF ± MEKi therapy (n = 461/5231, 8.8%) and subsequently showed a CR (n = 37/5231, 0.71%), and assessed their clinical course. 

### Statistical Analysis

Descriptive statistics were used to analyze the baseline characteristics of the study population. Treatment duration was calculated as the period between initial drug administration and the date of treatment discontinuation. The duration of the initial response was calculated from the date of the first administration of BRAF ± MEKi to the date of tumor progression or last follow-up. We employed Kaplan-Meier plots to illustrate median overall survival (OS) and progression-free survival (PFS) probabilities and to calculate the median OS and PFS of the investigated groups. Survival curves were compared by using the log-rank test. We performed univariate and multivariate Cox regression analysis to evaluate the impact of certain clinical baseline characteristics (e.g., sex, age, or ulceration) on OS. In order to quantify the impact on PFS and OS, we used Hazard ratios (HR) with 95% confidence intervals (CI). In all cases, two-tailed *p*-values were calculated and considered significant with values *p* < 0.05. SPSS (version 23, IBM, Ehningen, Germany), RStudio (Version 1.3.959) and GraphPad PRISM (Version 5, GraphPad Software, San Diego, CA, USA) were used for all analyses. 

## 3. Results

### 3.1. Baseline Patient Characteristics

Among the 5231 melanoma patients in the ADOreg database, 2441 patients had an unresectable stage III or stage IV melanoma, and from these 461 patients (18.9%) received BRAFi ± MEKi treatment. Among patients receiving BRAF ± MEKi therapy, 37 patients (15 female, 22 male) developed a CR (8.0% of all advanced melanoma patients receiving BRAF ± MEKi therapy) and were enrolled into this retrospective study (Figure 1). 

For this cohort showing an initial CR, the maximum observation period starting from the first application of BRAF ± MEKi therapy comprised 100 months. The mean patient age at TT initiation was 58 years (interquartile range, IQR: 33–77 years). The median primary tumor thickness was 3.2 mm (IQR: 1.33–5.00 mm). Eleven patients (29% of all patients) had an ulcerated primary tumor. All patients had confirmed BRAF-V600-mutations in their melanoma and 24 patients (64.9%) showed a BRAF-V600E-mutation. The median time interval from primary cancer diagnosis to metastasis was 15.0 months (IQR 1.5–58.0 months), with 9 patients (24.3%) showing an AJCC stage IV upon initial presentation and 5 patients (13.5%) having been diagnosed with melanoma brain metastasis (MBM) at the time of initial presentation. During the observation period, patients developed metastasis in a median of 3 different metastatic sites (range: 0–7) and 11 patients developed MBM in the course of the disease (Table 1). 

All patients included in this retrospective study received BRAFi ± MEKi for at least one month and responded with an initial CR to TT as the BOR. All patients were naïve to systemic treatment, and TT with BRAFi ± MEKi was administered as first-line therapy. Overall, three patients received monotherapy with vemurafenib (2/37) or dabrafenib (1/37) and 34 patients received a combination of BRAFi and MEKi treatment, including five patients with encorafenib and binimetinib, 20 patients with dabrafenib and trametinib, and nine patients with cobimetinib and vemurafenib (Table 1). 

### 3.2. Duration of BRAF/MEK Inhibitor Therapy

The median duration of TT in the entire cohort was 16 months (IQR: 6.5–28.0 months) with 11 patients still receiving the initial therapy with BRAFi± MEKi at the time of data cut-off (median treatment duration: 25.0 months). Among patients who terminated TT, the most common causes for treatment cessation were tumor progression (n = 13/26, 50%) and toxicity (n = 6/26, 23.1%). In seven patients, treatment was discontinued due to patient preference or at recommendation of the physician (in the following collectively referred to as physicians’ choice) (27%). When comparing the duration of treatment for patients who terminated BRAF ± MEKi for various reasons we observed that patients terminating TT due to toxicity received BRAF ± MEKi for the shortest time interval (median 5.5 months) compared to patients who terminated therapy due to TP (median: 10.0 months) or physicians’ choice (median: 12.0 months) (Figure 2). The median observation period after treatment cessation was 19 months (range: 0–70 months). 

### 3.3. Treatment Duration Is Not Correlated with the Risk of TAE, But with a Longer PFS and OS

Our results revealed that 40% of patients recorded treatment-associated adverse events (TAE) of ≥CTCAE grade 1 during BRAF ± MEKi therapy including fever (11%), diarrhea (13%), or colitis (6%). Notably, the occurrence of TAE was not correlated with the duration of TT (*t*-test: 0.29). On the contrary, treatment-associated toxicities leading to discontinuation of therapy mainly occurred within the first 12 months of therapy (n = 5/6), thus reflecting the notion that patients terminating TT due to toxicity received BRAFi/MEKi for a shorter time period compared to patients ceasing TT for other reasons. 

### 3.4. Factors Associated with Disease Progression and Survival upon BRAF/MEKi Treatment

Different clinical and pathological factors are associated with disease progression in the early stage of melanoma, such as ulceration or tumor thickness [27,28]. Thus, we investigated whether established prognostic factors, such as T-status, Breslow-index, ulceration, the number of metastatic tumor sites, the duration of treatment, the presence of MBM or AJCC-stage, and baseline patient characteristics (i.e., gender, age) might impact the OS in patients with complete responders upon BRAFi ± MEKi, using a multivariate Cox-Model. Here, the number of metastatic sites, and the ulceration status were found to increase the risk of disease progression upon TT, whereas the duration of initial TT was found to decrease the risk of disease progression. In accordance, we could show that patients who received BRAF ± MEKi therapy for a longer time period than 16 months had a significantly longer PFS and OS compared to patients receiving BRAF ± MEKi therapy for less than 16 months (median PFS 46 months vs. 7 months, *p* < 0.001, Figure 3; median OS: 77 months vs. 36 months, *p* = 0.037, Appendix A). 

### 3.5. Treatment Outcomes upon BRAF/MEKi Therapy

Tumor progression after initial CR to BRAF ± MEKi therapy was found in 22/37 patients and was most common in patients who had to discontinue TT in the course of the disease. 19/37 patients (54%) exhibited a relapse while under BRAFi ± MEKi treatment, whereas 3/37 patients (8.1%) relapsed after termination of BRAFi ± MEKi. Among patients who terminated BRAFi ± MEKi treatment while in CR, the median time to recurrence following treatment cessation was 1 month (mean: 10.9 months, range: 0–28 months). When excluding those patients who had to discontinue TT due to adverse events the median time from treatment cessation to relapse was 28 months (mean: 17.3 months). The median duration of treatment cessation among the 26 patients who had terminated TT was 0 months, with 20/37 patients terminating TT for less than 6 months, and 6/37 terminating TT for a longer interval than 6 months. By contrast, eleven of the 37 patients with CR (30%) received TT throughout the observation period, and all of these maintained their CR. Overall 15/37 (40.5%) patients remained progression-free throughout the observation period. The median PFS for the overall cohort, as assessed by Kaplan-Meier method, was 27 months (95% CI: 1.5–52.5 months) and median OS was 77 months (95% CI: 19.0–135.0 months). 

### 3.6. Impact of Treatment Discontinuation

Investigating the impact of treatment discontinuation on the primary endpoints, we observed that patients who had discontinued TT (irrespective of the reasons) had a significantly shorter PFS (median PFS: 10 vs. not reached, *p* < 0.001, Figure 4) and OS compared to patients receiving ongoing TT (median OS: 77 months vs. not reached, *p* = 0.049, Appendix A). 

Due to the strong heterogeneity in terms of treatment and outcome within this subcohort (see Table 2), we further distinguished patients according to the reasons for TT cessation or the duration of TT cessation. Here, our data revealed that 9/13 patients (69.2%) who discontinued TT for other reasons than PD experienced tumor progression after a median of 8 months, whereas 4/13 patients (30.8%) remained tumor-free. Progression-free survivors were mainly identified among patients who discontinued TT due to the wish of the patient or physicians’ choice (75%). Therefore, we compared patients who had to discontinue TT due to TAE or PD (n = 6 and n = 13) with patients who discontinued therapy for other reasons (n = 7) and patients who received ongoing TT (n = 11).

Our results unveiled, that patients who discontinued treatment either due to toxicity or PD did have a significantly shorter PFS (median PFS 3 months vs. 10 months vs. 40 months vs. not reached, *p* < 0.0001) compared to patients having discontinued for other reasons or receiving ongoing TT (Figure 5). Moreover, it was found that patients who discontinued therapy after previously having received TT for at least 12 months had a significantly longer PFS as compared to patients ceasing TT after a short initial BRAF/MEKi therapy (median PFS: 40 months vs. 6 months, *p* < 0.0001; Figure 6). 

Furthermore, we could observe that the median PFS and OS of patients who had stopped TT for a period of at least 6 months did not statistically differ from patients receiving ongoing BRAF ± MEKi therapy (median PFS 43 months vs. not reached, *p* < 0.001; median OS not reached in both groups, *p* = 0.06), whereas patients who terminated BRAF ± MEKi therapy for less than 6 months had a significantly shorter PFS and OS (median PFS: 9 months; median OS: 36 months) (Appendix A). 

### 3.7. Duration of Response and Second-Line Treatment

Nearly 60% of patients (22/37) who initially responded with a CR to BRAFi ± MEKi treatment eventually showed TP at a later time point, requiring the application of second-line treatments that comprised either a re-induction of BRAFi ± MEKi (n = 5), initiation of ICI therapy (n = 15) or treatment with the oncolytic virus talimogen laherparepvec (n = 1) (Table 1). Among these patients receiving second-line treatments, 16 patients relapsed while under first-line BRAFi ± MEKi treatment, and 5 patients relapsed subsequent to BRAF ± MEKi termination. In patients that relapsed after discontinuation of TT, the median interval between discontinuation of first-line TT and the first administration of second-line ICI was 0 months (range: 0–13 months), whereas for patients with a re-induction of BRAF ± MEKi the median interval was 23 months (range: 0–28 months). Patients with second-line ICI received either a combination of ipilimumab (IPI) and nivolumab (Nivo) (33%) or monotherapy with Nivo (27%), IPI (13%), pembrolizumab (20%), or atezolizumab (7%). Notably, patients were less likely to respond to second-line ICI when ICI therapy was initiated subsequent to PD during BRAF ± MEKi (CR: n = 1, PR: n = none; ORR: 12.5%), as compared to patients who received ICI after discontinuation of BRAF ± MEKi for other reasons than PD (CR: n = 2, PR: n = 1; ORR: 42.8%).

Kaplan-Meier analysis including all patients receiving second-line ICI therapy revealed a median OS of 24 months and median PFS was 2 months (Appendix A). Overall, response to second-line ICI (defined as CR, PR or MR) was weak (n = 4/15; 27%) and CR could only be observed in 20% of patients (median PFS: 12 months). By contrast, three of the five patients who stopped first-line BRAFi ± MEKi, responded with a PR/CR (60%) to BRAFi ± MEKi-rechallenge, which lasted for a median of 9 months. Two of these patients remained relapse-free until the end of the observation period (median 9.5 months). Considering the widely divergent outcomes within the second-line ICI therapy and second-line TT cohorts, we hypothesized that certain clinical characteristics might help to identify patients with an increased risk of disease progression. Therefore, we performed logistic regression analysis to assess whether clinical characteristics correlate with the PFS of those two groups. In our study, we found no significant differences concerning T-stage (T2B vs. T2A), median AJCC stage at the time of initial diagnosis (IIIA vs. IIIA), median Breslow-thickness (3.6 mm vs. 3.15 mm), or the presence of tumor ulceration in the primary tumor (33% vs. 23%). 

## 4. Discussion

The advent of immune checkpoint inhibitors (e.g., anti-PD-L1, anti-PD-1, anti-CTLA4) and the targeted inhibition of the MAPK pathway with BRAF and MEK inhibitors in BRAF-V600-mutant melanoma patients has led to profound and durable tumor responses in some patients with advanced melanoma [5,6]. Recent data suggests an overall survival of more than 5 years and a PFS >20% after 3 years in a subset of patients who have shown complete response to initial BRAF ± MEKi treatment [3]. Although patients undergoing a CR thus have a favorable survival, relapses while on treatment occur in approximately 30% of patients [29]. 

To date, there are no reliable clinical factors or markers which might predict the long-term response of BRAFi/MEKi after an initial CR. Hence, unlike many immunotherapy studies, the treatment in BRAF/MEKi-trials is mostly continued indefinitely. However, it has been reported that BRAF/MEKi therapy might also affect the tumor microenvironment (TME) and improve durable tumor surveillance, thus having a long-term beneficial effect. It has been shown that these long-term effects are most likely in patients with CR to TT [30]. In particular, it has been suggested by Long and colleagues, that patients with a CR and favorable baseline characteristics, such as a lower initial tumor burden, fewer metastatic tumor sites, or normal LDH-levels, may be more likely to show a stronger and more sustained response, thus driving long-term survival [19]. Nonetheless, the issue of whether TT should be continued until tumor progression or whether it can be stopped at an earlier time point without posing a significant risk of disease progression in this particular subgroup of patients remains unsolved.

In this retrospective study, we present the outcomes of a series of 37 patients who obtained CR upon BRAF ± MEK inhibition. The median duration of initial TT has been 16 months, with 11 patients still receiving therapy at the time of data cut-off. We observed that 60% of patients with an initial CR eventually showed tumor progression. Notably, disease progression has only been observed in patients who ceased BRAF ± MEKi therapy after obtaining an initial CR. Aligning with these results, median PFS and OS were significantly worse in patients who had to discontinue TT. Patients with a durable CR after treatment cessation were predominantly identified in a subcohort of patients who received initial TT for a long time period, who discontinued TT for other reasons than PD or toxicity, and who discontinued TT for a longer interval than 6 months (n = 4/6). Interestingly, baseline patient characteristics, including the median time of patient follow-up, have been similar for patients receiving ongoing TT. 

These findings are consistent with previous reports investigating the outcome after treatment cessation in patients with a CR to initial BRAF ± MEKi therapy (see Table 3). These reports revealed the occurrence of long-term and progression-free survivors after discontinuation of TT. However, rates of melanoma recurrence (0–50%), median intervals to recurrence subsequent to TT discontinuation (3.0–6.6 months), and median follow-up after treatment cessation (12–19 months) showed strong variations depending on the specific trial [2,4,22,31,32,33]. Notably, all case reports reported high rates of treatment cessation due to toxicity (54%-100%), with the exception of Warburton and colleagues who excluded patients who terminated TT due to toxicity from their retrospective study [31]. 

Our cohort differs from those previously published studies since we have included patients who continuously received TT and patients who ceased TT for various reasons. Therefore, we were able to compare the outcome of all these different subgroups and thus provide further insight into whether and when BRAF ± MEKi therapy can be safely discontinued in complete responders. In particular, patients included in our retrospective study received BRAF ± MEKi therapy for a relatively long time period before ceasing TT. Only two studies have reported a longer initial treatment with BRAF/MEKi [31,33]. However, patients terminating BRAF ± MEKi in this retrospective study for other reasons than PD showed a higher rate of tumor progression (69.2%) and a shorter PFS upon termination of TT, which might be explained by the longer overall observation period covered in our study [22,34,35]. 

In contrast to a previous report, we could further observe a strong correlation between the duration of the initial BRAF ± MEKi treatment and the maintenance of response after treatment cessation [22]. In particular, our analysis revealed that patients who received TT for a longer period than 12 months prior to treatment discontinuation had a significantly longer PFS as compared to patients with a shorter initial treatment. Therefore, we suggest that the duration of previous BRAF ± MEKi treatment might impact the strength and durability of the response after treatment discontinuation. Considering the off-target effects of BRAF/MEKi therapy, such as a paradoxical activation of CD8+ T-cells, it seems conceivable that a longer initial treatment with TT might prime anti-tumor immunity towards a more durable response even after treatment cessation [36]. However, due to the small number of patients included in our retrospective study, the clinical significance of our results certainly requires confirmation in larger prospective trials. Moreover, and consistent with previous trials, we could confirm that patients showing positive prognostic markers, such as a smaller number of metastatic tumor sites, no melanoma brain metastases and a smaller tumor burden at the time of initial diagnosis are more likely to undergo a prolonged response to BRAF ± MEKi therapy even after treatment discontinuation. 

In most clinical trials with BRAF ± MEKi therapy, the main reason for discontinuation of treatment was disease progression [4,22]. In our retrospective study, disease progression has also been the main reason for treatment cessation (n = 13/22). Disease progression was largely found within the first 12 months of treatment (n = 9/13). By contrast, only two patients who had to discontinue therapy due to other reasons had shown a disease progression prior to 12 months of therapy (n = 2/7). 

An important observation of our study is the correlation between the initial duration of BRAF ± MEKi therapy and sustained response durability upon discontinuation of TT. Moreover, we could show that discontinuation of TT due to disease progression (n = 13) or toxicity (n = 6) occurred at early time points after the initial administration of TT. In contrast, patients who terminated TT for other reasons had received BRAFi/MEKi for a longer period. Consistent with these findings, patients terminating BRAF/MEKi therapy for other reasons than PD or toxicity showed a significantly longer PFS and OS. 

Although TT resulted in an initial CR in all patients investigated, tumor progression was observed in nearly 60% of patients, mostly affecting patients who had to discontinue TT. One common explanation for melanoma recurrence after an initial CR and subsequent discontinuation of treatment are micrometastases, which remain under cytostatic control during TT, while again proliferating upon treatment cessation [22]. However, this hypothesis might not explain the prolonged progression-free survival after treatment cessation, which has been found in some patients. In this regard, it has been suggested that BRAF/MEKi therapy—next to its cytostatic effects—might also contribute to tumor control via immunomodulatory effects, such as increasing melanoma immunogenicity, paradoxical activation of effector T cells, or reducing the infiltration of tumor-associated macrophages and myeloid-derived suppressor cells [36,37]. 

Our data regarding tumor progression are consistent with those of larger clinical trials. Notably, patients which have discontinued TT in this study had a higher risk of disease progression (n = 9/13) compared to a previously published study (n = 3/12) [31]. In our study tumor recurrence resulted in an alteration of the anti-tumor regiment and administration of ICI. Response to ICI in the second-line setting varies from 15.4% in case of pembrolizumab to 12% for combined IPI + Nivo treatment [38]. After DC-vaccination [39] or first-line ICI [6] response rates have been reported to be much higher (35–72%). By contrast, median PFS after progression upon combined TT has been reported to be 2.6 months for anti-PD-1 monotherapy vs. 2.0 months for a combination of anti-PD-1 and anti-CTLA-4 therapy. Objective response rates were found in 18.0% and 15.0%, respectively, whereas median OS has been found to be 8.4 months for nivolumab treatment and 7.2 months for combined Ipi + Nivo treatment [40]. Notably, these reports have included all patients receiving BRAF ± MEKi treatment. 

In our study second-line ICI was administered in 15 patients subsequent to disease progression. The median interval between the initial application of first-line BRAF/MEKi therapy and the first administration of ICI was 10 months. Overall response rate (27%) and median PFS (2 months) to ICI in this second-line setting were poor as compared to ORR and median PFS in treatment-naïve patients [6]. This observation is consistent with the narrative of an aggressive melanoma progression once TT is discontinued [41]. While certain clinical factors such as the frequent appearance of MBM, a higher tumor burden, and increased levels of LDH are known factors that dampen the response to subsequent ICI [42,43] recent molecular studies have underlined the immunogenic profile of TT and its shift towards an immune inhibitory tumor microenvironment once secondary resistance occurs. This leads to a decrease in CD8+ T cell infiltration and an increase of regulatory T cells [44,45]. Furthermore, decreased PD-L1 expression levels at progress have been observed [46,47]. Hence, this shift in the TME at time of tumor progression might potentially contribute to the weak response found in patients receiving second-line ICI.

Reactivation of the MAPK signaling pathway is largely responsible for an acquired resistance against BRAF/MEKi therapy either due to emerging new spliced variants of BRAF, development NRAS mutation, or loss of feedback inhibition leading to BRAF dimerization resulting in resistance to BRAFi [48,49,50]. Nonetheless, it has been reported, that the rechallenge of BRAF ± MEKi therapy in patients who initially responded to TT is associated with a further response: Here, Valpione and coworkers have reported an objective response rate of 42% and PFS of 5 months after BRAFi rechallenge in 116 patients, which is consistent with the results of our study (ORR: 60% and PFS: 9 months) and previous publications [2,35,51]. Determining the optimal individual treatment after disease progression upon initial BRAF ± MEKi therapy might therefore be another important issue for clinical trials, as it may contribute to a better outcome and even lead to durable tumor responses in melanoma patients. 

When interpreting the results of our analysis, limitations to be considered are the retrospective nature and the small cohort of patients under investigation, which may limit the significance and the accuracy to predict patients who may obtain long-term benefits of the treatment regimen. Therefore, interpretation of subgroup analysis in particular requires caution. Nonetheless, the multi-institutional approach may help overcome bias from a single-center analysis. Also, our patient collective features certain clinical characteristics that are consistent with those of larger studies. 

In summary, we could provide evidence for the occurrence of patients who have maintained an initial CR even after cessation of TT. Our data indicate that patients who have initially received at least 16 months of TT, and who did not have to discontinue TT due to PD or toxicity are more likely to obtain a long-term response. The observation that treatment cessation due to toxicity is associated with a short PFS further suggests that treatment discontinuation due to toxicity should be avoided and toxicity control should be considered an important element of BRAF/MEK-directed TT not least with respect to the patients’ quality of life. The retrospective design of our study, however, does not allow for a reliable prediction of patients who will obtain long-term benefit and with no prospective data available to guide treatment decisions, stopping BRAF/MEKi therapy in CR cannot be recommended. Thus, prospective randomized clinical trials involving a tightly monitored discontinuation of TT with further translational end-points might further clarify the issue for whom it would be safe to cease treatment at which time point. Notably, recruitment and ethics of such study might be difficult since the occurrence of disease progression has been found to significantly worsen a subsequent reinduction of treatment. 

Based on our data, it might however be conceivable to conduct further studies on BRAF ± MEKi treatment cessation in patients who have initially obtained a CR, show good prognostic clinical features, who have received TT for at least 16 months without showing disease relapse and who have not discontinued treatment due to PD or treatment-associated toxicities. During the first 6 months of subsequent treatment discontinuation, we further suggest a close clinical monitoring allowing for the rechallenge of BRAF± MEKi therapy in case of disease recurrence. The data presented here should assist clinicians to make informed and individualized decisions, taking into consideration the risks and chances (e.g., avoidance of cumulative side effects, paradoxical secondary malignancies) of treatment cessation, while patients that do not fit the outlined criteria should remain on treatment, as our data has unequivocally confirmed that patients who continued TT have the longest PFS and OS. 

## Figures and Tables

**Figure 1 cancers-13-02312-f001:**
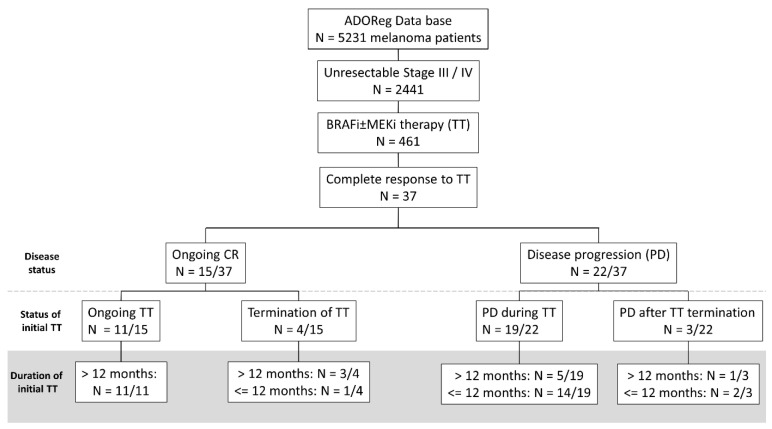
Flow chart showing the selection steps of the retrospective study and the treatment outcomes of melanoma patients after an initial CR to BRAF/MEKi therapy. We included all patients with an unresectable stage III / IV melanoma receiving BRAF/MEK-inhibitor (i) therapy who showed an initial complete response (CR) (n = 37). Among these 37 patients, 40.5% experienced an ongoing CR, whereas 59.5% showed disease progression during follow-up. 11 patients who showed an ongoing CR continuously received BRAF ± MEKi therapy, whereas 4/15 patients discontinued TT previously, but yet showed an ongoing CR. Notably, most patients with an ongoing CR received initial BRAF ± MEKi therapy for a longer interval than 12 months. By contrast, patients showing a disease progression after initial CR, received BRAF ± MEKi therapy for less than 12 months and most patients experienced disease progression (PD) during BRAF ± MEKi therapy.

**Figure 2 cancers-13-02312-f002:**
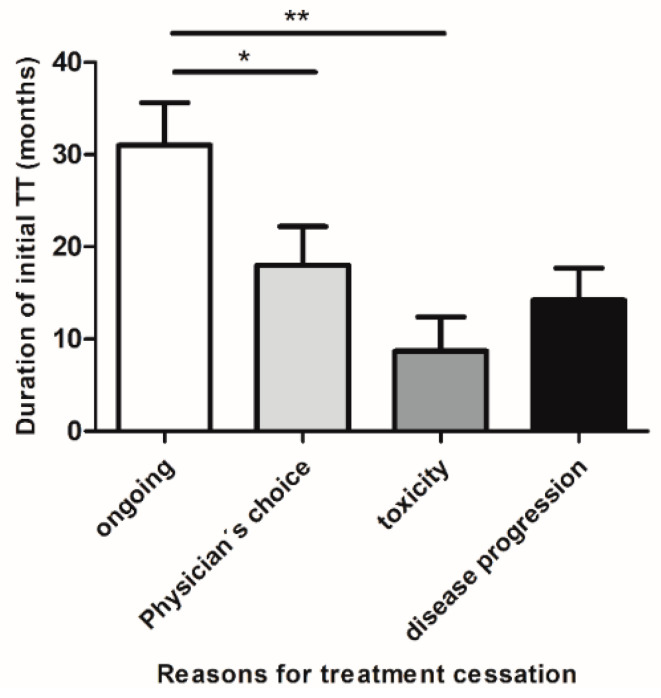
Duration of BRAF/MEKi therapy in patients who obtained CR and either maintained treatment or discontinued TT for various reasons. Patients who discontinued first-line BRAF ± MEKi treatment due to treatment-associated toxicities (mean duration ± SEM: 8.7 ± 3.7 months) or due to disease progression (mean ± SEM: 14.2 ± 3.4 months), received therapy for a shorter time period compared to patients who discontinued therapy for other reasons (mean ± SEM: 18 ± 4.2 months) i.e., patient preference or at recommendation of the treating physician (collectively termed: physician’s choice). Patients with ongoing treatment had received BRAF ± MEKi therapy for the longest time period (mean ± SEM: 31 ± 4.6 months). Abbreviations: ** p* < 0.05; *** p* < 0.005.

**Figure 3 cancers-13-02312-f003:**
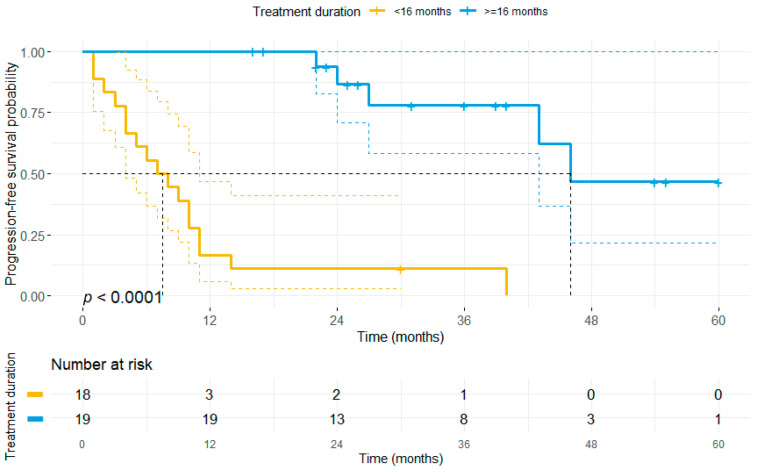
The duration of first-line BRAF ± MEKi therapy correlates with progression-free survival. The patient cohort was stratified based on the median duration of first-line BRAF± MEKi treatment (16 months) in the entire study population. Patients who received first-line BRAF ± MEKi therapy for more than 16 months showed a significantly longer PFS (median: 46 months) as compared to patients receiving TT for a shorter time period (median PFS: 7 months).

**Figure 4 cancers-13-02312-f004:**
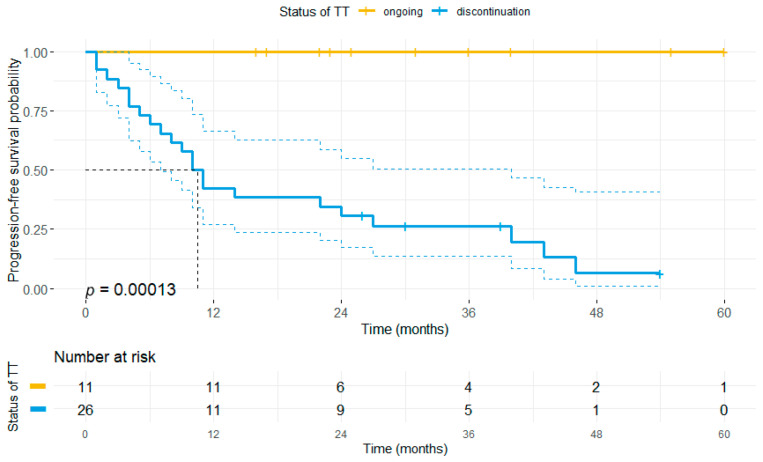
Discontinuation of TT correlates with the risk of disease progression. All patients who obtained CR upon BRAF ± MEKi treatment and received therapy until data cut-off (n = 11) maintained their CR, whereas 22 of 26 patients who discontinued BRAF ± MEKi therapy at any time point during the observation period eventually relapsed. Accordingly, patients who had terminated TT showed a significantly shorter PFS (median: 10 months) compared to patients with ongoing BRAF ± MEKi therapy. Notably, patients who maintained their CR until treatment cessation showed a lower risk of tumor relapse after treatment cessation (n = 3/7) as compared to all patients ceasing TT.

**Figure 5 cancers-13-02312-f005:**
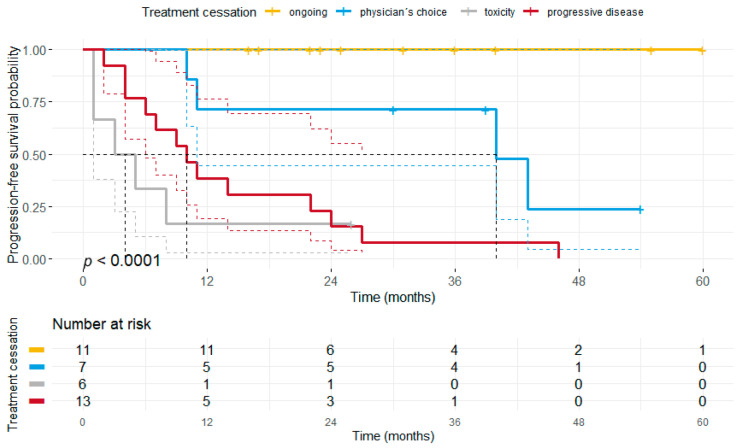
Kaplan-Meier plot illustrating the progression-free survival separated by the reasons for treatment discontinuation. Patients who terminated BRAF ± MEKi therapy due to toxicity or disease progression had a significantly shorter PFS (median PFS: 3 months vs. 10 months) compared to patients with ongoing treatment (median PFS not reached, *p* < 0.0001) or treatment discontinuation due to other reasons (termed as physician’s choice) (median PFS: 40 months, *p* = 0.013).

**Figure 6 cancers-13-02312-f006:**
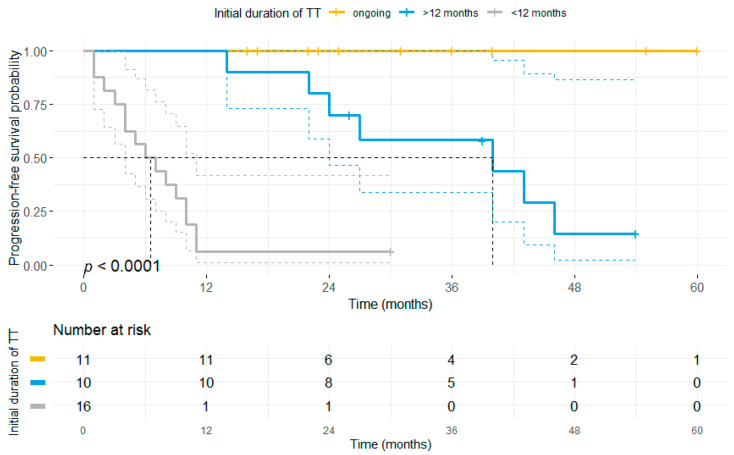
Initial duration of BRAF ± MEKi therapy correlates with progression-free survival even after discontinuation of TT. The patient cohort was stratified based on the status of BRAF ± MEKi therapy (ongoing, yellow vs. discontinuation: blue and grey) and the initial duration of first-line BRAF ± MEKi treatment. Patients who received first-line BRAF ± MEKi therapy for more than 12 months prior to treatment discontinuation (blue) showed a significantly longer PFS (median: 40 months) as compared to patients receiving TT for a shorter time period (median PFS: 6 months) prior to treatment cessation.

**Table 1 cancers-13-02312-t001:** Baseline patient characteristics.

Clinicopathological Features	N (%)
**Median age** at initiation of TT (range)	58 (29–80)
Female	15/37 (40.5%)
Male	22/37 (59.5%)
**Primary tumor**	
Median Breslow thickness (range)	3.2 mm (0.5–30.0 mm) ^1^
Ulceration	11/28 (39.3%) ^2^
**Clinical course prior to initiation of targeted therapy**	
Median interval from primary diagnosis to advanced stage melanoma (range)	15 months (0–285 months)
Median time from primary diagnosis to initiation of BRAF ± MEKi therapy (range)	21 months (0–290 months)
**Metastatic lesion**	
**Metastatic sites, median (range)**	3 (0–7)
Lung	25/37 (67.6%)
Liver	6/37 (16.2%)
Nodal	18/37 (48.6%)
Cutaneous	7/37 (18.9%)
Bone	7/37 (18.9%)
Other	17/37 (45.9%)
Cerebral	16/37 (43.2%)
**ECOG ^3^-status ≥ 1 at beginning of TT**	4/16 (25%)
**Treatments**	
Treatment with BRAF ± MEKi (first-line) ▪ Vemurafenib ▪ Dabrafenib ▪ Encorafenib plus Binimetinib ▪ Dabrafenib plus Trametinib ▪ Cobimetinib plus Vemurafenib	2/37 (5.4%) 1/37 (2.7%) 5/37 (13.5%) 20/37 (54.1%) 9/37 (24.3%)
Median treatment duration (range) ▪ Ongoing treatment (n = 11) ▪ Cessation of therapy for other reasons than PD or toxicity (n = 7) ▪ Treatment cessation due to PD or toxicity (n = 19)	16.0 months (1–60 months) 25 months (16–60 months) 22 months (0–70 months) 3 months (0–49 months)
Treatment-related adverse events of any grade ▪ Discontinuation of BRAF ± MEKi-treatment due to toxicity	15/37 (40.5%) 6/37 (18.9%)
Tumor progression	22/37 (60%)
Median progression-free survival (95% CI) in months	27.0 months (1.5–52.5)
**Second-line treatment** ▪ **Immune-Checkpoint blockade** ○ Median time interval between discontinuation and ICB-induction (range) ○ Response to second-line ICB ○ Median progression-free survival upon ICB ▪ **Reinduction of BRAF ± MEKi therapy** ○ Median interval between discontinuation and rechallange (range) ○ Response to second-line BRAF ± MEKi therapy ○ Median progression-free survival	21/37 (56.7%) 0 months (0–13 months) 4/15 (27%) 2 months 23 months (0–28 months) 3/5 (60%) 9.5 months
**Follow-up**	
Overall observation period upon TT initation	100 months
Median follow-up after TT discontinuation (range)	19 months (0–70 months) ^4^
Median overall survival (95% CI) in months	77 months (19–135 months)
Deceased	10/37 (27.0%)

**Abbreviations**: ^1,2,3^ Statistics based on the total number of patients with known Breslow thickness (n = 28), ulceration status (n = 28) and Eastern Cooperative Oncology Group (ECOG)-status (n = 16); ^4^ Statistics based on the total number of patients (n = 26) who have discontinued BRAF/MEKi therapy; PD = progressive disease; ICB = immune-checkpoint blockade; ICI = immune-checkpoint-inhibitors; TT = targeted therapy; TAE = treatment-associated adverse events; Response was defined as CR and PR.

**Table 2 cancers-13-02312-t002:** Patient characteristics in a subcohort of patients who discontinued TT due to reasons other than PD.

No.	Age	Sex	Initial AJCC Stage	MBM	Mutation	Therapy	TT (mo)	Reason for Cessation	Cessation Time (mo)	Time to Relapse after Cessation (mo)	TP	BOR to 2nd Line Therapy	Status	OS (mo)
1	45	M	IB	no	V600E	Vem	30	Physician’s choice	13	56	yes	PD (IPI)	AWD	100
2	48	M	IIB	no	V600E	Dab + Tram	30	Physician’s choice	9	-	no	-	AWD	39
3	79	F	IV	yes	V600R	Cob + Vem	29	Physician’s choice	24	-	no	NA (Cob + Vem)	AWD	54
4	75	M	IIIB	yes	V600E	Dab + Tram	5	Physician’s choice	24	-	no	-	AWD	30
5	55	M	IIA	no	V600E	Dab + Tram	12	Physician’s choice	28	19	yes	PR (Dab + Tram)	AWD	59
6	48	M	IIIC	yes	V600E	Dab	11	Toxicity	0	35	yes	CR (Tram)	DC	36
7	71	M	IV	no	V600K	Dab + Tram	11	Physician’s choice	0	22	yes	MR (Nivo)	AWD	34
8	59	M	IV	yes	V600E	Cob + Vem	5	Toxicity	0	14	yes	PD (IPI + Nivo)	AWD	20
9	45	M	IV	yes	V600E	Dab + Tram	3	Toxicity	0	27	yes	PD (Nivo)	AWD	30
10	33	F	IIIC	no	V600E	Cob + Vem	9	Physician’s choice	0	48	yes	CR (IPI)	AWD	59
11	74	M	IIIB	no	V600K	Cob + Vem	6	Toxicity	1	2	yes	PD (Atezo)	DC	10
12	61	M	IB	no	V600E	Dab + Tram	26	Toxicity	0	-	no	-	AWD	26
13	55	F	IIA	yes	V600K	Enco + Bini	1	Toxicity	0	19	yes	CR (Nivo)	AWD	20

**Ab****breviations:** Pat = Patient; M = male; F= female; mo= months; MBM= melanoma brain metastases; Vem = Vemurafenib; Dab = Dabrafenib; Tram = Trametinib; Cob = Cobimetinib; Bini = Binimetinib; Enco = Encorafenib; TT = targeted therapy; Physician’s choice = comprises wish of the patient or physician’s recommendation; PD = progressive disease; TP= tumor progression; BOR = best overall response; MR = mixed response; IPI = Ipilimumab; Nivo = Nivolumab; Atezo = Atezolizumab; AWD = alive with disease, DC = deceased.

**Table 3 cancers-13-02312-t003:** Comparison of outcomes reported for metastatic melanoma cohorts with an initial CR to BRAFi ± MEKi therapy.

Reports Analzying Outcomes after TT Cessation	Number of Patients (CR)	Median Duration of TT Treatment (Months)	Discontinuation due to Toxicity	Median Follow-up after Discontinuation (Months)	Tumor Progression (%)	Median PFS upon TT Cessation	Response to TT Rechallange
Warburton [31]	13	39	0%	19	0%	5	100%
Wyluda [34]	3	12	100%	15	0%	NA	NA
Desvignes [32]	6	6	100%	15	100%	4	17%
Vanhaecke [33]	16	21	63%	12	53%	2.5	63%
Tolk [35]	12	13	54%	17	46%	3	50%
Carlino [22]	12	NA	100%	16	50%	6.6	33%
Stege	37	16	16%	19	69%	1	60%

## Data Availability

All relevant data are within the manuscript and its Appendix A. The retrospective data used for statistics have been collected within the framework of the ADOReg.

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
