# Peer review of "Discontinuation of BRAF/MEK-Directed Targeted Therapy after Complete Remission of Metastatic Melanoma—A Retrospective Multicenter ADOReg Study"

_cancers, 2021, doi:10.3390/cancers13102312_

Round 1

Reviewer 1 Report

Stege and Haist et al. performed a retrospective statistical analysis on patients' data undergone BRAF/MEK inhibitor targeted therapy to shed further light on the optimum duration of treatment with this combination therapy. The authors investigated the correlation of the targeted therapy with survival and risk of disease progression and made some interesting conclusions (backed up by their data analytics) regarding the efficacy of BRAFi/MEKi based on its applied duration.

The intro and results are comprehensive and easy to read, and the methodology and analytics are reasonably defined, but some arguments need to be touched upon more in the conclusion section of the manuscript. For example, the notion that "Therefore, we suggest that the duration of previous BRAF±MEKi treatment might impact the strength and durability of the response after treatment discontinuation." is not very well explained. How can your specific suggestions affect the treatment paradigm for these patients? The study has the potential to offer stronger insights backed up by data presented in this research.

Minor Miswordings:

Line 364: inside > insight

Author Response

1. Stege and Haist et al. performed a retrospective statistical analysis on patients' data undergone BRAF/MEK inhibitor targeted therapy to shed further light on the optimum duration of treatment with this combination therapy. The authors investigated the correlation of the targeted therapy with survival and risk of disease progression and made some interesting conclusions (backed up by their data analytics) regarding the efficacy of BRAFi/MEKi based on its applied duration.

The intro and results are comprehensive and easy to read, and the methodology and analytics are reasonably defined, but some arguments need to be touched upon more in the conclusion section of the manuscript. For example, the notion that "Therefore, we suggest that the duration of previous BRAF±MEKi treatment might impact the strength and durability of the response after treatment discontinuation." is not very well explained. How can your specific suggestions affect the treatment paradigm for these patients? The study has the potential to offer stronger insights backed up by data presented in this research.

Ad 1: We would like to thank the reviewer for the valuable criticism and comments on our research article, which we have taken into account in our revised manuscript. Particularly referring to the correlation found between the initial duration of BRAF/MEKi therapy and the response durability after treatment cessation, we have now explained and discussed our observations in a more precise manner in order to specify the suggestions drawn thereof (Page 13, Line 377-382). Due to the small number of patients included in our retrospective study, we do however admit that the clinical significance of our conclusions may be limited and require further confirmation in prospective studies. Considering the pathophysiological mechanisms resulting from treatment with BRAF/MEKi, such as a paradoxical activation of CD8+ CTL, and higher immunogenicity of melanoma tumor cells, it seems conceivable that a longer initial treatment of BRAF/MEKi might prime anti-tumor immunity towards a more durable response even after treatment cessation.

2. Minor Miswordings:

Line 364: inside > insight

Ad 2: We thank the reviewer for this important hint. Therefore, we have corrected the miswording reported by the reviewer (Page 12 Line 365). Furthermore, our manuscript has undergone a comprehensive English style review in order to provide a precise and smooth English therein. 

Please see attachment for a comprehensive summary of all point-to-point answers to the reviewers´ comments.

Reviewer 2 Report

I congratulate the authors, this is a well written very important work, I have only minor suggestions

Page 2 line 93 - BOR first mentiones give abbrevation here (not on the next page), page 3 line 1 ICI please give abbrevation here and not in the abstract (there used only once), 

Results page 4 - line 152, please correct the % statistics you write 2441 patients had an unresectable [..] melanoma, and from these 461 (18.8% not 8.8%!) received..., further line 153 please omit the 0.71% as it is not relevant

page 7/8 figure 3 or 4: one of the figures can be omitted - as most treatment discontinuations were for tumor progression this seems a little redundant, you could include one of the supplement files in the main manuscript instead

page 7 line 218 - MBM please specify abbrevation

page 9 table 2: please provide abbreviation for MR and again page 14 line 423 TME (please provide abbreviation)

Author Response

  1. I congratulate the authors, this is a well written very important work, I have only minor suggestions: Page 2 line 93 - BOR first mentiones give abbrevation here (not on the next page), page 3 line 1 ICI please give abbrevation here and not in the abstract (there used only once)
  1. Results page 4 - line 152, please correct the % statistics you write 2441 patients had an unresectable [..] melanoma, and from these 461 (18.8% not 8.8%!) received..., further line 153 please omit the 0.71% as it is not relevant
  2. page 7/8 figure 3 or 4: one of the figures can be omitted - as most treatment discontinuations were for tumor progression this seems a little redundant, you could include one of the supplement files in the main manuscript instead
  3. page 7 line 218 - MBM please specify abbrevation
  4. page 9 table 2: please provide abbreviation for MR and again page 14 line 423 TME (please provide abbreviation)

Ad 1, 4, and 5: We thank the reviewer for the kind assessment of our research and have taken into account the valuable criticism. Thus, we have provided the relevant abbreviations at the positions indicated: (i) BOR, best overall response, we have included the abbreviation on Page 2 Line 93 and removed it from the Patients and Methods section; (ii) ICI, immune-checkpoint-inhibitor, the abbreviation was first given in the introduction (Page 2 Line 75/76), therefore we have refrained from revising the indicated abbreviation in Line 96; (iii) MBM, melanoma brain metastases, we have specified the indicated abbreviation on Page 5 Line 173; (iv) MR, mixed response, we have included the aforementioned abbreviation in the Table legends on Page 9 Line 271; (v) TME, tumor microenvironment, the indicated abbreviation has previously been specified (Page 11 Line 336).

Ad 2: We agree with the reviewer that the % statistics should provide a more focused description for the cohort of patients with advanced, unresectable melanoma. To this end, we have revised the indicated % statistics and provided the relative numbers for advanced melanoma patients who have (i) received BRAF/MEKi therapy (18,9% of all unresectable stage III or IV melanoma patients) and (ii) who have obtained CR upon BRAF/MEKi therapy (8,0% of all advanced melanoma patients receiving BRAF/MEKi therapy and 1,5% of all patients with unresectable, stage III or IV melanoma); Page 4 Line 152-154.

Ad 3: We agree with the reviewer that Figures 3 and 4 might – at least in parts – display redundant results, which might be attributed to the share of patients ceasing TT due to tumor progression. Despite this partial redundancy we are convinced that both figures give valuable insights into factors impacting the outcome of patients receiving BRAF/MEK-directed TT. In particular, we could show that both the duration of BRAF/MEKi therapy and the status of BRAF/MEKi therapy were strongly correlated with progression-free and overall survival, even when excluding patients that ceased TT due to tumor progression (median PFS for patients receiving TT>16 months: not reached vs. TT<16 months: 8,0 months, p<0,001; median PFS for patients ceasing TT for other reasons than tumor progression: 11 months vs patients receiving ongoing treatment: not reached, p=0,002; not shown). The observation that the duration of BRAF/MEKi therapy is correlated with progression-free survival might particularly be relevant given the finding that the duration of initial BRAF/MEKi therapy also impacted the PFS of patients after discontinuing TT. In order to provide a more distinct demarcation between the two figures, we have however taken into account the reviewer´s valuable criticism and added a more concise and thorough description of our observations in the figure legends (L255-257). 

Reviewer 3 Report

Topic: Discontinuation of BRAF/MEK-directed targeted therapy after complete remission of metastatic melanoma – a retrospective multicenter ADOReg study

------------------------------------------------

The authors retrospectively analyzed selected data on patients with BRAF-V600 advanced melanoma and CR upon BRAF-MEK-directed TT, who were included into the multicentric skin cancer registry ADOReg of the Dermatologic Cooperative Oncology Group (DeCOG) until the data cut-off (04/2020).  

Data on the age at the start of TT with BRAFi and MEKi, sex, primary localization of melanoma, site of  metastasis, LDH levels in serum, systemic pretreatments, duration of treatment, best overall response  (BOR) to treatment, duration of the initial response, overall status of the patient, reasons for discontinuation, subsequent course of the disease and second-line treatment were collected. BOR was defined as complete response (CR), partial response (PR), stable disease (SD), or progressive disease (PD). PD was defined by disease recurrence at any site during observation according to standard RECIST criteria. In this retrospective analysis we included patients with advanced non-resectable melanoma (n=2441/5231, 46.7%) who received first-line BRAF±MEKi therapy (n=461/5231, 8.8%) and subsequently showed a CR (n=37/5231, 0.71%), and assessed their clinical course.

Of 461 patients who received BRAF/MEK-directed TT 37 achieved a CR. TP after initial CR was observed in 22 patients (60%) mainly affecting patients who discontinued TT (n=22/26), whereas all patients with ongoing TT (n=11) maintained their CR. Accordingly, patients who discontinued TT had a higher risk of TP compared to patients with ongoing treatment (p<0,001). However, our data also show that patients  who received TT for more than 16 months and who discontinued TT for other reasons than TP or toxicity   did not have a shorter PFS compared to patients with ongoing treatment

Main comments:

  1. Objectives of the study were achieved
  2. Impressive, detailed and thorough analysis of the data
  3. Major limitations – small cohort of patients investigated which impacts the clinical significance of the conclusion of the study

                               – retrospective nature - impacts the prediction potential patients who may obtain long-term benefits of the treatment regimen with no prospective data for guidance.

  1. Major strengths: - thoroughness and depth of the data analysis

                               - the multi-institutional involvement of the investigation; helps limit probable bias.

  1. Message to clinicians: is rather too non-committal                                   

Author Response

  1. Topic: Discontinuation of BRAF/MEK-directed targeted therapy after complete remission of metastatic melanoma – a retrospective multicenter ADOReg stud. The authors retrospectively analyzed selected data on patients with BRAF-V600 advanced melanoma and CR upon BRAF-MEK-directed TT, who were included into the multicentric skin cancer registry ADOReg of the Dermatologic Cooperative Oncology Group (DeCOG) until the data cut-off (04/2020). Data on the age at the start of TT with BRAFi and MEKi, sex, primary localization of melanoma, site of  metastasis, LDH levels in serum, systemic pretreatments, duration of treatment, best overall response  (BOR) to treatment, duration of the initial response, overall status of the patient, reasons for discontinuation, subsequent course of the disease and second-line treatment were collected. BOR was defined as complete response (CR), partial response (PR), stable disease (SD), or progressive disease (PD). PD was defined by disease recurrence at any site during observation according to standard RECIST criteria. In this retrospective analysis we included patients with advanced non-resectable melanoma (n=2441/5231, 46.7%) who received first-line BRAF±MEKi therapy (n=461/5231, 8.8%) and subsequently showed a CR (n=37/5231, 0.71%), and assessed their clinical course.Of 461 patients who received BRAF/MEK-directed TT 37 achieved a CR. TP after initial CR was observed in 22 patients (60%) mainly affecting patients who discontinued TT (n=22/26), whereas all patients with ongoing TT (n=11) maintained their CR. Accordingly, patients who discontinued TT had a higher risk of TP compared to patients with ongoing treatment (p<0,001). However, our data also show that patients  who received TT for more than 16 months and who discontinued TT for other reasons than TP or toxicity   did not have a shorter PFS compared to patients with ongoing treatment

Main comments:

  1. Objectives of the study were achieved
  2. Impressive, detailed and thorough analysis of the data
  3. Major limitations – small cohort of patients investigated which impacts the clinical significance of the conclusion of the study– retrospective nature - impacts the prediction potential patients who may obtain long-term benefits of the treatment regimen with no prospective data for guidance.
  1. Major strengths: - thoroughness and depth of the data analysis

                               - the multi-institutional involvement of the investigation; helps limit probable bias.

  1. Message to clinicians: is rather too non-committal         

We thank the reviewer for the kind assessment of our research article and the comments, which we have considered in our revised manuscript. We have particularly taken into account the valuable comments concerning the major limitations of our study, which include the small cohort of patients and the retrospective nature of the study which impact both the clinical significance of our conclusion and the strength of clinical markers to predict patients who may obtain long-term benefits of the treatment regimen given. Therefore, we have provided a more concise description of our study´s limitations and the resulting impact on our conclusions (Page 14 Line 437-439).

Reviewer 4 Report

This is an excellent paper of considerable clinical relevance and would be an important addition to the literature. It is very well presented, with clear and pertinent text, tables and figures.

Throughout, I hoped that a parallel study had been ongoing, to improve understanding of the variable responses the authors report and help explain the reasons affecting treatment efficacy, including those for disease progression or increased treatment toxicity in some patients. The obvious ones that immediately come to mind are whether clonal evolution, with further genetic aberrations, or suppressed adaptive immune function, which the authors allude to in their Discussion, or even both, are the major determinants.  Such a parallel study, if not already underway, should be considered seriously for future prospective studies. In the meantime, this largely retrospective observational study is very well designed and presented. I congratulate the authors on an excellent paper.

Very minor points: spacing between paragraphs is not standardised on my version. Also, very occasionally abbreviations are used without annotating the full term first and, conversely, very occasionally the full term is used later in the script following earlier explanation of the abbreviation. eg line 93, BOR used but I couldn't find any previous mention - perhaps I missed it! I was amused to find 44 different definitions of this acronym when I searched the internet, so best to define it.

Author Response

  1. This is an excellent paper of considerable clinical relevance and would be an important addition to the literature. It is very well presented, with clear and pertinent text, tables and figures.Throughout, I hoped that a parallel study had been ongoing, to improve understanding of the variable responses the authors report and help explain the reasons affecting treatment efficacy, including those for disease progression or increased treatment toxicity in some patients. The obvious ones that immediately come to mind are whether clonal evolution, with further genetic aberrations, or suppressed adaptive immune function, which the authors allude to in their Discussion, or even both, are the major determinants.  Such a parallel study, if not already underway, should be considered seriously for future prospective studies. In the meantime, this largely retrospective observational study is very well designed and presented. I congratulate the authors on an excellent paper. Very minor points: spacing between paragraphs is not standardised on my version. Also, very occasionally abbreviations are used without annotating the full term first and, conversely, very occasionally the full term is used later in the script following earlier explanation of the abbreviation. eg line 93, BOR used but I couldn't find any previous mention - perhaps I missed it! I was amused to find 44 different definitions of this acronym when I searched the internet, so best to define it.

We thank the reviewer for the kind assessment of our study and are happy to provide novel insights into understanding the variable responses after treatment cessation of BRAF/MEK-directed targeted therapy. Indeed, it would be highly interesting to conduct a prospective analysis of patients who have obtained CR upon BRAF/MEK-directed targeted therapy and either continuously received TT or ceased TT after a predefined time interval. This approach might particularly enable the identification of potential clinical characteristics which might predict the patients´ long-term response after BRAF/MEKi therapy discontinuation. However, recruitment and ethics of such prospective studies might be difficult due to the worsened prognosis after tumor relapse. Therefore, current considerations in our laboratory mainly involve a more comprehensive retrospective analysis comprising a larger cohort of patients in order to generate a solid data basis prior to the initiation of further prospective studies.  

Furthermore, we thank the author for the important minor points raised in the comments: In our revised manuscript we have therefore provided a standardized spacing between the paragraphs (and annotated the full term of the abbreviations where it has first been used in the manuscript (i.e., BOR has been specified as best overall response in Line 93).
